# Exogenous Application of Coconut Water to Promote Growth and Increase the Yield, Bioactive Compounds, and Antioxidant Activity for *Hericium erinaceus* Cultivation

**Preuk Chutimanukul** [1,2,*] , **Siripong Sukdee** [1] , **Onmanee Prajuabjinda** [3] , **Ornprapa Thepsilvisut** [1,2] , **Sumalee Panthong** [3] , **Hiroshi Ehara** [4] and **Panita Chutimanukul** [5]

[1] Department of Agricultural Technology, Faculty of Science and Technology, Thammasat University, Pathumthani 12120, Thailand
[2] Center of Excellence in Agriculture Innovation Centre through Supply Chain and Value Chain, Thammasat University, Pathumthani 12120, Thailand
[3] Department of Applied Thai Traditional Medicine, Faculty of Medicine, Thammasat University, Pathumthani 12120, Thailand
[4] International Center for Research and Education in Agriculture, Nagoya University, Nagoya 464-8601, Japan; ehara@agr.nagoya-u.ac.jp
[5] National Center for Genetic Engineering and Biotechnology (BIOTEC), National Science and Technology Development Agency, Khlong Luang, Pathumthani 12120, Thailand
* Correspondence: preuk59@tu.ac.th

**Abstract:** *Hericium erinaceus* (Bull.:Fr) Pers. is a medicinal mushroom that has various health benefits and is a rich source of bioactive compounds and antioxidant activity. In recent years, *H. erinaceus* has been considered for its many medicinal properties and is widely consumed in Asian countries. This work aimed to explore the potential improvement of coconut water utilization in order to promote growth and increase the yield, as well as the enhancement of bioactive compounds and antioxidant activity of *H. erinaceus*. The application of coconut water at a concentration of 20% ($v/v$) resulted in the optimal development and production of *H. erinaceus*. In terms of biological efficiency, it was demonstrated that the 22.09% increase over the control treatment resulted in a higher yield. Moreover, *H. erinaceus* cultivated from coconut water intake at all concentrations resulted in higher protein content. Additionally, bioactive compounds such as total triterpenoid content and total phenolic content of *H. erinaceus* were 67.87–89.24 mg Urs/g DW and 16.62–17.39 mg GAE/g DW, respectively, while *H. erinaceus* grown on a control treatment had the lowest at 56.84 mg Urs/g DW and 14.42 mg GAE/g DW, respectively. Applying coconut water at all concentrations resulted in *H. erinaceus* $IC_{50}$ values of 0.58–0.69 mg/mL exhibiting higher DPPH activities than those grown on control treatment $IC_{50}$ value of 0.77 mg/mL. Therefore, this finding of the study indicated that the utilization of coconut water as a therapy resulted in a significant enhancement in the development, production, and bioactive compounds of *H. erinaceus*, in comparison to the control treatment.

**Keywords:** *Hericium erinaceus*; coconut water; yield; bioactive compounds; antioxidant activity

## 1. Introduction

Mushrooms are generally consumed for their nutritional values and are a good source of nutrients and health-promoting compounds [1,2]. The consumption of mushrooms has garnered significant interest in academic circles due to their distinct and refined taste, as well as their abundance of minerals and notable nutritional worth, particularly in terms of protein content [3–5]. Mushroom fruit bodies have the potential to serve as valuable sources of nutritional content and bioactive chemicals, which have been shown to possess significant health benefits [6,7]. Currently, the identification of over 2000 species of mushrooms that possess edible or therapeutic properties has been accomplished [8,9].

The species *Hericium erinaceus* (Bull.:Fr) Pers., also known as Yamabushitake, Lion's Mane, or Houtou, is a member of the family Hericiaceae, order Russulales, phylum Basidiomycota, is considered to be economically significant and valuable. As a fungal organism possessing renowned health-enhancing attributes, the medicinal mushroom is widely recognized [10,11]. This mushroom has attracted considerable attention for various medicinal properties [12]. Numerous beneficial actions have been demonstrated by several isolated substances derived from *H. erinaceus*. Several physiologically active substances have been isolated from *H. erinaceus*, which have been found to exhibit diverse biological activities. Furthermore, the presence of distinct bioactive chemicals in these extracts has been scientifically confirmed [11]. As a result, it is crucial to consider the advantages of *H. erinaceus* concerning its potential utilization in functional food and biomedical products. This is because *H. erinaceus* contains various nutritive components, including high-quality protein, dietary fibers, essential vitamins, and minerals. Additionally, it serves as a valuable source of bioactive compounds that contribute to overall well-being, such as polysaccharides, hericenone, erinacine, phenolic compounds and triterpenoid [13–15]. Thus, *H. erinaceus* has garnered considerable interest among biological researchers because of its advantageous medicinal attributes, including its ability to facilitate wound healing, exhibit anticancer qualities, serve as an antioxidant, lower blood pressure, regulate blood sugar levels, modulate the immune system, and perhaps treat neurodegenerative disorders, among other therapeutic possibilities [16,17]. Species under the genus *Hericium* are recognized for their suitability as edible mushrooms, as well as their utilization in the production of medicinal extracts and dietary supplements [18–20]. With the increasing attention to the values of mushrooms, mushroom production has also gradually increased worldwide. The most widely cultivated around the world are edible mushrooms [21,22]. In order to improve the yield and quality of medicinal mushrooms, various cultivated strategies were employed.

The exogenous application of coconut water is a potential alternative approach for the advancement of *H. erinaceus* culture techniques. This is due to the presence of organic substances within coconut water, namely phytohormones such as cytokinin, auxin, and gibberellins, which have been seen to exert a stimulatory influence on cell division [23]. This is due to the discovery of fungal Histidine kinases (HKs) are among the most prominent sensing proteins studied in the kingdom Fungi, which have a high degree of similarity with plant hormone receptors. Phytohormones are known to play crucial roles in plant development, and recent works have highlighted cytokinin interaction at diverse levels of biosynthetic and metabolic pathways. Also, coconut water, with its many applications, is among the important natural products that are safe for users and influence environmental sustainability [24]. There is a growing body of scientific data supporting the utilization of specific substances such as phytohormones or plant hormones as growth supplements or growth accelerators in plants [23]. From previous reports on the mycelial biomass production of Lentinus native isolates by Dulay et al. [22], it was found that the use of coconut water was the most suitable liquid culture medium for the maximum mycelial biomass production of all studied Lentinus, with special biological properties from nature. Therefore, the main objective of this research is to examine the possible advantages of administering coconut water in order to augment the yield, bioactive compounds, and antioxidant activity of *H. erinaceus*. Additionally, the study aims to contribute to the diffusion of information, thereby facilitating the application of these findings in the improvement of other mushroom cultivation methods.

## 2. Materials and Methods

### 2.1. Chemicals and Reagents

High-quality chemicals, including 98% sulphuric acid, 99.8% acetic acid glacial, and 95% nitric acid, were acquired from RCI Labscan Limited (Bangkok, Thailand), petroleum ether 40–60 °C were obtained from Qrec (Chonburi, Thailand), 70–72% perchloric acid were obtained from Sigma-Aldrich (St. Louis, MI, USA), Kjeldahl copper catalyst tablets were obtained from Oskon (Samut Prakan, Thailand), 81.0–83.0% ammonium molyb-

date ($(NH_4)_6Mo_7O_{24}\cdot4H_2O$), 99% ammonium metavanadate ($NH_4VO_3$), 99.5% sodium carbonate ($Na_2CO_3$), 97.0% Sodium hydroxide (NaOH), 85% Potassium hydroxide (KOH), 99% potassium chloride (KCl) and 99% potassium dihydrogen phosphate ($KH_2PO_4$) were obtained from KemAus (Bangkok, Thailand), and vanadate-molybdate reagents were obtained from Ricca Chemical (Arlington, TX, USA). The reagents and standards of 99% vanillin (4-Hydroxy-3-methoxybenzaldehyde), 90% ursolic acid (3β-hydroxy-urs-12-en-28-oic acid), Folin–Ciocalteu's reagent, 98% gallic acid (3,4,5-Trihydroxybenzoic acid), DPPH (1,1-diphenyl-2-picrylhydrazyl), and 98% butylated hydroxytoluene (2,6-Di-tert-butyl-4-methylphenol) were acquired from Sigma-Aldrich (St. Louis, MI, USA). The organic solvents used in this study were 99.8% ethyl alcohol absolute ($C_2H_5OH$) and 99.8% methanol ($CH_3OH$), both obtained from the Merck company (Darmstadt, Germany).

### 2.2. Analyses of the Physicochemical Properties and Nutrient Content of H. erinaceus Substrate before Cultivation

The physicochemical characteristics and nutrient content of the substrate were assessed prior to cultivation. This involved the preparation of the substrate by oven drying at a temperature of 70 °C for a duration of 72 h and was achieved and sieved through a 2 mm sieve to analyze specific substrate physicochemical properties, such as pH, using a pH meter PC950, Apera Instrument (Columbus, OH, USA). For electrical conductivity (EC), measured using a conductivity meter Eutech CON 2700, Thermo Fisher Scientific (Waltham, MA, USA). Moisture content was determined according to Horwitz [25]. Additionally, organic carbon (OC) was analyzed using a CHNS/O Analyzer model 628 series, Leco Corporation (St. Joseph, MI, USA), organic matter (OM) (calculated by organic carbon × 1.724) and carbon to nitrogen ratio (C:N ratio) (calculated by dividing the organic carbon value by nitrogen).

The present study used a modified method based on the approach developed by Chutimanukul et al. [26] for the analysis of nutrient content, which consisted of the analysis of total nitrogen (N), total phosphorus (P), and total potassium (K). The nitrogen in the substrate was analyzed by using an elemental analyzer CHNS/O Analyzer, phosphorus (P) was analyzed by using a UV-Spectrophotometer UV-1280, Shimadzu (Kyoto, Japan) at the wavelength of 882 nm, and potassium (K) by using a flame photometer 410C, Sherwood Scientific Ltd. (Cambridge, UK).

### 2.3. Substrate Preparation and Mushroom Cultivation

In December 2022, specimens of *H. erinaceus* were grown for the purpose of investigation in Pathumthani, Thailand (13°59′30.2″ N 100°38′18.2″ E). The cultivation substrate comprises rubberwood sawdust and soybean meal at the ratio 76:4. Each of these sawdust substrates was then mixed with corn cob and rice bran at the ratio 12:3:1 on a dry weight basis, then mixed well and adjusted the humidity with water to 80 percent. For mushroom cultivation, 750 g of each substrate were placed into plastic bottles. Subsequently, the sample was subjected to autoclaving at a temperature of 121 °C for a duration of 30 min, while maintaining a pressure of 15 pounds per square inch. Following autoclaving, the sample was allowed to equilibrate at ambient temperature for a period of 24 h. The *H. erinaceus* inoculum was subsequently introduced into the bottle containing the substrate. The plastic bottles, which contained the inoculated substrate, were then placed into a greenhouse where the environmental conditions were regulated to maintain a temperature of $20 \pm 2$ °C and a relative humidity of $80 \pm 5\%$. This was carried out to facilitate the colonization of mycelium.

### 2.4. Study of Growth and Yield of H. erinaceus with Coconut Water

The procedure of preparing coconut water involves the careful selection of young coconuts (*Cocos nucifera* L.) that have not undergone any processing. Additionally, aged coconuts, namely those that are 170–200 days after pollination (DAP), are chosen for this purpose. These coconuts are sourced from the planting sites located in Damnoen

Saduak, Ratchaburi Province, Thailand. Subsequently, an analysis was conducted on coconut water, specifically focusing on the phytohormones. The levels of phytohormones in coconut water was determined by capillary electrophoresis-tandem mass spectrometry (CE-MS/MS). All CE-MS experiments were performed with an Agilent capillary electrophoresis system in conjunction with an Agilent Trap XCT mass spectrometer equipped with an Agilent CE-ESI-MS sprayer kit (G1607A) and an Agilent CE-ESI-MS adapter kit (G1603A), Agilent Technologies (Waldbronn, Germany). Analysis was conducted according to the experimental method of Tan et al. [27]. The concentration of the coconut water was then modified using sterilized distilled water, resulting in five distinct degrees of concentration: 0 (serving as the control), 20, 40, 60, 80, and 100% (*v/v*).

In the process of *H. erinaceus* cultivation, when it reached the fruiting body opening stage, 4 mL (by using an auto pipette to measure into the spray bottle) of coconut water was sprayed to touch the spawning surface once after the fruiting body opening process. Upon the completion of the harvesting period, each experimental mushroom was collected and characterized in the following manner: Number of *H. erinaceus* caps in each bottle (cap), and the quantification of caps in each inoculum was performed according to the method of Chutimanukul et al. [26] by segregating the mushrooms at the stalk junctions and calculating the mean number of caps obtained as growth values (Figure 1). Then, the diameter of the fruit bodies of *H. erinaceus* (centimeter) was measured using a vernier caliper. For determination of yield, fruit bodies of *H. erinaceus* were harvested and the obtained yield was measured as the fresh weight and dry weight (in grams). The calculation of biological efficiency (%) was performed using the following equation:

$$\text{Biological efficiency (\%)} = \frac{\text{Fresh weight of mushrooms (g)}}{\text{Dry weight of substrate after harvest (g)}} \times 100 \qquad (1)$$

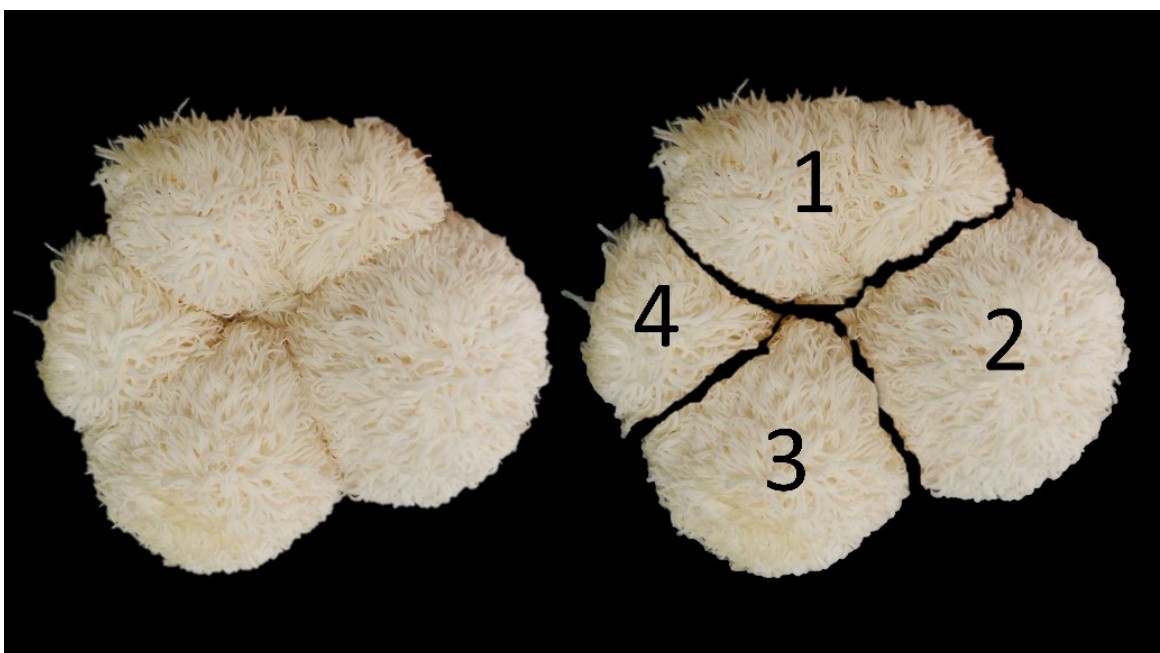

**Figure 1.** The classification of fruit bodies at the stalk junction is depicted in the figure, whereby numerical values 1–4 correspond to the number of caps.

## 2.5. Nutritional Composition of H. erinaceus

Fresh fruit bodies were subjected to oven drying at a temperature of 65 °C for a duration of 72 h. Subsequently, the dried samples were pulverized into a fine powder using a commercially available blender, in order to analyze nutrients. The moisture content, crude protein content, crude fat content, dietary fiber, ash content, and total carbohydrate was

assessed using the methodologies outlined by the Horwitz [25]. The crude protein content (Total nitrogen × 4.38) of the samples was estimated by the macro-Kjeldahl method by using a Kjeldahl digestion system–KjelDigester K-449, BUCHI (Flawil, Switzerland). The crude fat was determined by extracting 1 g of powder sample with 150 mL of petroleum ether, using a Gerhardt Soxtherm (Gerhardt, Germany), and the ash content was determined by incineration at 550 °C for a minimum of 4 h or overnight until obtaining white ash.

The dietary fiber content was analyzed by means of a Gerhardt Fibretherm Laboratory Instrument FT12 (Gerhardt, Germany). The analytical procedure relies on boiling an accurately weighed amount of the air-dried powdered sample with sulfuric acid, washing with water till free from acidity, followed by boiling the residue with potassium hydroxide and rinsing with water till free from alkalinity. The non-soluble residue was then dried (at 105 °C), weighed and incinerated (at 550 °C) until ash is formed. The difference between the ash content and the non-soluble residue in relation to the weight of the initial ample represents the dietary fiber content.

Total ash content accurately weighed amount of 1 g dried *H. erinaceus* powder was heated in a Muffle furnace (CARBOLITE, 1100 °C, Chamber Furnace, ELF models, England) at 550 °C for 4 h to determine a value for ash. The crucible containing the total ash was allowed to cool in a desiccator till constant weight, weighed and the total ash calculated by difference.

The calculation of total carbs was performed using the following equation:

$$\text{Total carbohydrates} = 100 - \text{Percentage of (moisture + protein + fat + ash + fiber)} \quad (2)$$

The energy was calculated based on the equation [5]:

$$\text{Energy} = 4 \times (\text{protein + carbohydrate}) + 9 \times (\text{fat}) \quad (3)$$

### 2.6. Preparation of H. erinaceus Ethanol Extract

The ethanol extraction process was conducted as outlined by Chutimanukul et al. [26]. The fresh fruit bodies of *H. erinaceus* were oven-dried at 65 °C for 48 h, then ground into powder and stored in sealed containers before analysis. Subsequently, a quantity of 10 g of mushroom powder derived from oven-dried fruit bodies of *H. erinaceus* was homogenized with 50 mL of 95% ethanol (*w/v*) at room temperature. Extracts were filtered through Whatman® Grade 1 qualitative filter paper. The process of maceration, including the extraction of substances, was carried out repeatedly every 3 days for a total duration of 9 days. The solvent-containing extract was dried under vacuum conditions using a rotary evaporator at the conclusion of the experiment (Rotavapor® R-300, BUCHI, Flawil, Switzerland). The crude ethanol extract was stored at 4 °C until use for bioactive compound and antioxidant activity analysis.

### 2.7. Total Triterpenoid Content Analyses of H. erinaceus

Total triterpenoid content in *H. erinaceus* extracts was quantified using a modified version of the technique developed by Ni et al. [28]. A portion of the extracts, measuring 300 μL, was transferred to a test tube. Afterward, 50 μL vanillin-acetic acid solution (5 mg mL$^{-1}$) and 800 μL perchloric acid were added, and the mixture was incubated for 15 min at 60 °C in a water bath. The mixture solution was then moved into an ice water bath. In the last phase, a volume of 5 mL of acetic acid was introduced into the mixture, which was afterwards allowed to stand undisturbed for a duration of 15 min at room temperature. Using a blank solution as a reference, the absorbance at a wavelength of 548 nm was measured using a microplate reader Multiskan GO, Thermo Scientific (Waltham, MA, USA). The quantification of triterpenoid content was performed by converting the measurements into milligrams of ursolic acid equivalents (Urs) per gram of dry weight (mg Urs/g DW). This conversion was achieved using a calibration curve constructed using standard solutions of ursolic acid (concentration of 1000, 500, 250, 120, 62.5, 31.25, and 16.625 μg mL$^{-1}$).

*2.8. Total Phenolic Contents Analyses of H. erinaceus*

Total phenolic contents in the *H. erinaceus* extracts were determined using the Folin–Ciocalteu method [29] with some modifications from Miliauskas et al. [30] The crude extract stock was prepared by dissolving in ethyl alcohol absolute. Subsequently, 20 μL of *H. erinaceus* extracts was combined with 100 μL of Folin–Ciocalteu's reagent (diluted at a ratio of 1:10), followed by an additional 80 μL of 7.5% sodium carbonate. Following a 30 min incubation period, the absorption at a wavelength of 765 nm was quantified using a microplate reader. The quantification of total phenol content was conducted by converting the measurements into milligrams of gallic acid equivalents (GAE) per gram of dry weight (mg GAE/g DW). This conversion was achieved by utilizing a calibration curve generated from gallic acid standard solutions with concentrations of 200, 160, 80, 40, 20, 10, and 5 μg mL$^{-1}$.

*2.9. Radical Scavenging Activity*

The scavenging potential of *H. erinaceus* extracts towards the radical 2,2-diphenyl-1-picrylhydrazyl (DPPH) was assessed using the methodology outlined by Seephonkai et al. [31] The extracts of *H. erinaceus* were dissolved in ethanol at concentrations of 3, 2, 1, 0.5, and 0.25 mg mL$^{-1}$. Subsequently, 100 μL of each sample was combined with 100 μL of a $6 \times 10^{-5}$ M DPPH solution (dissolved in 100% ethanol) in a 96-well microplate. Following a 30 min incubation period in a light-restricted environment at ambient temperature, the optical density was determined at a wavelength of 520 nm. The percentage of inhibition of the DPPH activity was determined by employing the below calculation:

$$\text{Inhibition (\%)} = [(A_{control} - A_{sample})/A_{control}] \times 100 \tag{4}$$

where $A_{control}$ is the absorbance of the DPPH solution, and $A_{sample}$ is the absorbance of the solution containing the sample.

The radical scavenging activity from DPPH radical scavenging assay was expressed as $IC_{50}$ value, which is the inhibitory concentration at which 50% of radicals were scavenged. The study was conducted with GraphPad PRISM 8.0.1. A low $IC_{50}$ value indicates the strong activity of the sample. BHT was employed as the positive control, whereas the negative control consisted of the lack of *H. erinaceus* extract.

*2.10. Statistical Analysis*

The study was performed in a completely randomized design (CRD), with 5 biological replicates per treatment. Each biological replicate contains 40 mycelium bottles. The experimental data were analyzed using a one-way analysis of variance with Duncan's multiple range test. The observed differences are deemed to have statistical significance when the *p*-value is less than 0.05. The statistical analyses were conducted using IBM SPSS Statistics 21.

**3. Results and Discussion**

*3.1. Analyses of the Physicochemical Properties and Nutrient Content of H. erinaceus Substrate before Cultivation*

The pH of the culture substrates utilized in the experiment exhibited physicochemical properties with a value of 6.77. The analysis of substrate pH before to spawning was shown to have an impact on the growth and development of mushrooms, consistent with the findings published by Imtiaj et al. [32], *H. erinaceus* exhibits optimal growth when cultivated on a substrate with a pH range of 5 to 9. Evidently, the pH level of the substrate utilized in mushroom growing fell inside the suitable range conducive to the proliferation of mushroom mycelium. The remarkable agronomic behavior observed can be attributed to the low electrical conductivity (EC) of the substrate, which was measured to be 1.39 dS m$^{-1}$. For a typical mushroom, optimal growth and development of the mycelium can be achieved at an EC around 0.87–1.98 dS m$^{-1}$ [33]. The study observed a moisture content of 69.2%

in the substrates, which aligns well with the appropriate moisture content required for growth. Additionally, a quantitative examination of the substrate revealed organic matter and organic carbon percentages of 46.00% and 79.30%, respectively (Table 1).

**Table 1.** Physicochemical properties and nutrient content of *H. erinaceus* substrate for cultivation.

| Constituents of Substrate | Contents |
|---|---|
| pH | 6.77 |
| EC (dS m$^{-1}$) | 1.39 |
| Moisture content (%) | 69.2 |
| Organic matter (%) | 79.30 |
| Organic carbon (%) | 46.00 |
| Total nitrogen (%) | 1.16 |
| Total phosphorus (%) | 0.10 |
| Total potassium (%) | 0.61 |
| C:N ratio | 39.66 |

Analysis of nutrients contained in the substrate found that nitrogen is an important nutrient for the growth effectively of *H. erinaceus* with a volume of 1.16%. This is the amount within the range that is favorable for the growth and development of *H. erinaceus* mycelium and provides an essential source of energy for the formation of mushroom cell structures [34]. Including analysis of the content of total phosphorus and total potassium of the substrate before spawning were 0.10 and 0.61%, respectively. However, phosphorus and potassium in the substrate are nutrients used for mushroom growth, although in small amounts, it still affects the *H. erinaceus* mycelium, it helps the physiological processes of mycelium grow normally. In addition, when considering the C:N ratio of the substrate, it was at a 39.66 ratio (Table 1), in which the C:N ratio is an important factor affecting the utilization of the nutrients from these substrates and mycelial growth in mushroom cultures [35]. Therefore, it is necessary to achieve a proper nitrogen and carbon balance in order to generate a C:N ratio ranging from 35 to 55, which is considered optimal for the successful cultivation of mushrooms [36]. Organisms affiliated with the kingdom of fungi exhibit significant variation in several areas of their structure and functioning. The capacity of microorganisms derived from mushroom mycelium to degrade cellulose, and lignin is a notable phenomenon. Therefore, they assume a significant function in the breakdown of organic substances to facilitate their use in the processes of development and photosynthesis for the production of bioactive molecules [37].

### 3.2. Phytohormones of Coconut Water

As the CE-MS/MS method provided high sensitivity as well as good accuracy, and precision during the validation procedure, the major challenge was to apply this method to screen for naturally occurring different classes of phytohormones in coconut water [27].

The analysis of coconut water in the experiment was performed under optimized conditions. Under the optimum CE-MS/MS conditions, the presence of phytohormones is a group of naturally occurring organic compounds that play crucial roles in regulating plant growth in a wide range of developmental processes. Several phytohormones that affect the growth of *H. erinaceus* are auxin, cytokinin, and gibberellin. From the analysis of plant growth regulators in coconut water contains have an auxin content of 138.9 μg mL$^{-1}$, various cytokinin content from 9.1 to 68.9 μg mL$^{-1}$, and gibberellins content from 13.9 to 33.4 μg mL$^{-1}$ (Table 2). The extensive use of coconut water as a growth-promoting component because it also appears to have growth regulatory properties, e.g., cytokinin-type activity. Additionally, the presence of auxin and gibberellins is implicated in many regulatory processes in mushrooms, especially those relating to the growth and development of mushrooms.

**Table 2.** Analysis of phytohormones in coconut water before use in the experiment.

|  | **Phytohormones Type** | **Contents in Coconut Water ($\mu$g mL$^{-1}$)** |
| --- | --- | --- |
| Auxin | indole-3-acetic acid | 138.9 |
| Cytokinin | dihydrozeatin *O*-glucoside | 40.2 |
|  | *trans*-zeatin *O*-glucoside | 42.8 |
|  | *trans*-zeatin riboside | 68.9 |
|  | *trans*-zeatin riboside-5$'$-monophosphate | 9.1 |
| Gibberellins | gibberellin 1 | 13.9 |
|  | gibberellin 3 | 33.4 |

### 3.3. Study on Growth and Yield of H. erinaceus

The results of *H. erinaceus* growth show that the cultivation substrate is the most important intrinsic factor. It is the sole source of essential nutrients necessary for the growth and development of mushrooms. The application of coconut water in the substrate helps the mushrooms catalyze the digestion of complex organic compounds in the substrate by releasing enzymes and then absorbing the small nutrients through the mycelia. Thus, it is able to stimulate the growth and productivity of *H. erinaceus* very well. As a result, there was a statistically significant increase in the number of cap per bottle and the diameter of the mushroom from the control treatment, by application of coconut water at a concentration of 20% (*v/v*), resulting in *H. erinaceus* the average number of cap was 5.6 cap (Table 3), and from measuring the diameter of cap (Figure 2A–F). It was found that coconut water treatment at 20–100% (*v/v*) concentration resulted in an average fruit bodies diameter of 9.02 to 9.83 cm, while the control treatment had an average fruit bodies diameter of 8.42 cm (Table 3). Nevertheless, the experimental findings indicated that the application of coconut water at a concentration of 20% (*v/v*) effectively promoted the development of *H. erinaceus*. This might be attributed to the presence of phytohormones in coconut water, which act as stimulants rather than nutrients. It is possible to employ limited quantities of the substance in order to facilitate and regulate the growth of mushrooms, including cellular and tissue metamorphosis. The mycelium of mushrooms has the ability to facilitate the transportation of phytohormones from coconut water by means of phytohormone receptors. In accordance with the notable discovery made by Hérivaux et al. [24], the examination of phytohormone receptors in fungi has identified fungal Histidine kinases that bear a significant resemblance to plant hormone receptors, specifically cytokinin receptors. This fungal receptor exhibits its functionality by responding to alterations in growth patterns subsequent to the reception of coconut water. Phytohormones are known to play a significant role in the growth of mushrooms, and recent studies have emphasized the involvement of cytokinins in several biosynthetic and metabolic processes. Previous studies have provided evidence supporting the notion that coconut water has a significant role in promoting the development of *H. erinaceus*. The observed enhancement in growth can be ascribed to the presence of auxins, cytokinins, and gibberellins, as these phytohormones are crucial regulators of plant growth and development [23,38]. Cytokinins have been extensively studied as important signaling molecules in the context of fungal biotic interactions [24,39]. Significantly, these homologs of phytohormone receptors are present not only in early evolving fungi that exhibit symbiotic or endophytic behavior towards plant roots, but also in early diverging fungal species that inhabit decomposing plant matter. Histidine kinases are highly prevalent sensory proteins that are found in fungi. Upon activation, such as in response to an external stimulus, Histidine kinases initiate phosphorylation cascades of varying complexity, ranging from two-component systems to multistep phosphorelays (in fungi). These cascades ultimately result in an adapted response that is involved in the morphogenesis of mushroom fruit bodies [24].

**Table 3.** Growth and yield of *H. erinaceus* obtained from various concentrations of coconut water.

| Concentration (%v/v) | Number of Cap (Cap) | Diameter of Cap (cm) | Fresh Weight (g) | Dry Weight (g) | Biological Efficiency (%) |
|---|---|---|---|---|---|
| 0 (control) | 3.8 ± 0.45 [c] | 8.42 ± 0.14 [c] | 72.79 ± 3.05 [d] | 8.26 ± 0.27 [c] | 31.51 ± 1.32 [d] |
| 20 | 5.6 ± 0.74 [a] | 9.83 ± 0.21 [a] | 88.86 ± 2.90 [a] | 10.07 ± 0.73 [a] | 38.47 ± 1.26 [a] |
| 40 | 4.5 ± 0.35 [b] | 9.64 ± 0.50 [a] | 78.40 ± 1.78 [c] | 9.02 ± 0.10 [b] | 33.94 ± 0.77 [c] |
| 60 | 4.6 ± 0.42 [b] | 9.45 ± 0.74 [ab] | 82.99 ± 3.09 [bc] | 9.18 ± 0.53 [b] | 36.14 ± 1.34 [bc] |
| 80 | 4.8 ± 0.57 [b] | 9.03 ± 0.18 [b] | 83.47 ± 5.03 [b] | 9.49 ± 0.40 [ab] | 35.92 ± 2.18 [bc] |
| 100 | 4.8 ± 0.57 [b] | 9.02 ± 0.39 [b] | 86.84 ± 4.50 [ab] | 9.94 ± 0.65 [a] | 38.11 ± 2.54 [ab] |
| F-test | ** | ** | ** | ** | ** |
| C.V.% | 10.95 | 3.83 | 4.11 | 6.91 | 4.36 |

Data are represented as mean ± SD (*n* = 5). Different letters in the same column indicate significant differences between the treatments according to DMRT at *p* < 0.05. ** There were significant differences at *p* < 0.01.

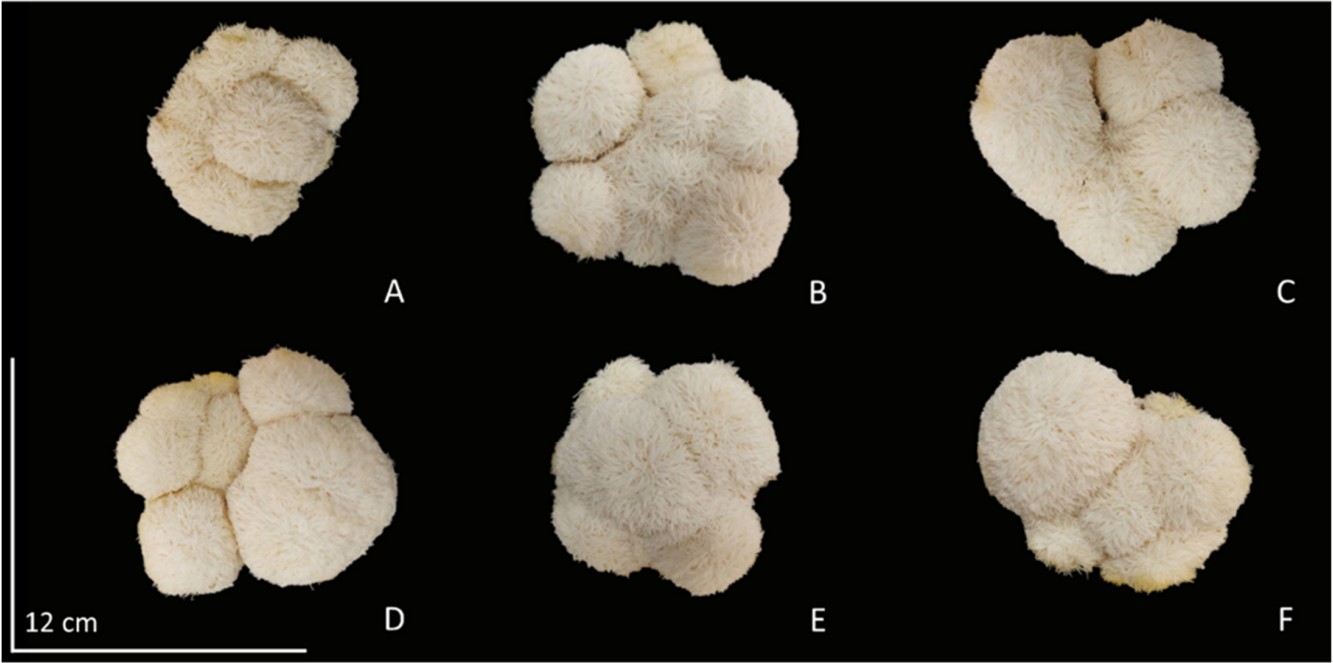

**Figure 2.** Diameter of *H. erinaceus* on various concentrations of coconut water (**A**) 0% (control), (**B**) 20%, (**C**) 40%, (**D**) 60%, (**E**) 80%, and (**F**) 100%.

The results of application coconut water on *H. erinaceus* production, adding coconut water to the cultures significantly increased the yield performance from the control treatment produced by the fruit bodies of *H. erinaceus*. For mushroom quantity, the fresh weight and dry weight after exposure by application of coconut water at a concentration of 20–100% (*v/v*), found that *H. erinaceus* had a fresh weight between 78.40 and 88.86 g, while the control treatment gave the least fresh weight, 72.79 g. The dry weight of *H. erinaceus* was also examined, which is the weight of the organic mass that grows into mushrooms and has been completely dehumidified by heat until the dry weight tends to be in the same direction as the fresh weight. The coconut water at a concentration of 20–100% (*v/v*) resulted in a dry weight between 9.02 and 10.07 g, while the control treatment gave the least dry weight of 8.26 g. In addition, considering the biological efficiency of *H. erinaceus* after all concentrations of coconut water were significantly different. Adding coconut water at a concentration of 20–100% (*v/v*) resulted in biological efficiencies between 33.94 and 38.47%, while the control treatment had the lowest biological efficiency of 31.51% (Table 3). From the results of this study, it may be caused by coconut water being a factor that can stimulate growth well, thus affecting the yield and biological efficiency of *H. erinaceus*, as well as coconut water will serve to regulate all vital activities cell division, and meristem formation

of *H. erinaceus* affecting growth, which is in line with the results of the experiment showing that the addition of coconut water may be another important factor affecting growth and yield. In addition, coconut water is an important factor in the uptake of macro- and micronutrients from the substrate constant for better *H. erinaceus* growth, and as Mustafa [40] reports, it was found that adding GA increases nitrogen usage efficiency by increasing the uptake of nitrogen from the substrate through nitrate reductase activity, which is required for nitrate uptake activity in the cells of fungi [41]. It can also be encouraging to the degrading ability of ligninolytic peroxidase enzymes produced by the mushroom mycelium to help in decomposing the sawdust for the accessibility of the nutrient present to be used for growth and development as a yield of *H. erinaceus* [42]. One of the important roles of phytohormones found in coconut water is that it acts as a regulator of morphological changes in fungi by tissue formation and cell division stimulation. This will affect the growth of the mycelium until it gathers into a fruit body mushroom. Therefore, the experiment of giving coconut water and transport generates phytohormones within mushroom mycelium that are a factor in the diverse regulation of various mushroom developmental processes.

*3.4. Nutritional Composition of H. erinaceus*

The results of the nutritional composition obtained for the studied *H. erinaceus* after receiving coconut water. Moisture contents that ranged between 6.07 and 6.29% were not a statistically significant difference. Since moisture content is an indication of the water content in mushrooms, the administration of coconut water as a growth regulator, which plays an important role in regulating cell proliferation and differentiation, has no effect on the moisture content of *H. erinaceus*. In the ash content of fruit bodies, the highest ash content was observed for coconut water at a concentration of 20–100%, which had an ash content between 11.71 and 12.88%, while the control treatment had the lowest ash content, measuring at 9.58%. However, these values are higher than those reported by Heleno et al. [5], where the ash content of *H. erinaceus* was 9.31%. The results showed that coconut water administration resulted in an increase in the ash content of *H. erinaceus*, which may be due to the mushroom mycelium being able to absorb more vitamins and minerals contained in coconut water during mushroom growth. In addition, the application of coconut water resulted in significantly higher crude protein content as compared to the control treatment. The application of coconut water at a concentration of 20–100% (*v/v*) found that *H. erinaceus* had a crude protein content between 24.99 and 27.74%, while the control treatment had a crude protein content of 23.48%. The increase in crude protein content was due to changes in nitrogen metabolism after receiving coconut water. Similarly, the crude fat contents of *H. erinaceus*, after receiving exposure to coconut water concentrations ranging from 20 to 100% (*v/v*), showed fat values between 3.03% and 3.40%. (Table 4). The nutritional composition of *H. erinaceus* observations revealed a different effect of the coconut water treatments. Data on the moisture content of the fruit body indicated no significant difference from the control. However, the increase in ash, protein, and fat levels resulting from the presence of coconut water plays an important role in the decomposition of organic matter by microorganisms. This decomposition process leads to the conversion of the organic matter into smaller molecules, which are subsequently utilized by cells as sources of energy and nitrogen for the synthesis of protein and fat [43] by using auxins and cytokinin in the coconut water to act as signaling molecules that transmit information regarding various environmental factors to the cell genome, which leads to the transformation of the response signal of the synthesis system protein and fat [44].

**Table 4.** Nutritional composition of *H. erinaceus* from the exogenous application of coconut water at various concentrations.

| Concentration (%v/v) | Nutritional Composition (%) | | | | | | Energy (kcal) |
|---|---|---|---|---|---|---|---|
| | Moisture Content | Ash Content | Crude Protein | Crude Fat | Total Carbohydrate | Dietary Fiber | |
| 0 | 6.29 ± 0.04 | 9.58 ± 0.35 [c] | 23.48 ± 0.82 [c] | 2.53 ± 0.31 [b] | 50.40 ± 0.85 [a] | 11.09 ± 0.96 [a] | 318.30 ± 3.16 [a] |
| 20 | 6.07 ± 0.10 | 12.33 ± 0.45 [ab] | 27.74 ± 0.41 [a] | 3.04 ± 0.54 [a] | 40.98 ± 1.31 [c] | 7.95 ± 0.17 [c] | 301.65 ± 3.94 [c] |
| 40 | 6.12 ± 0.19 | 11.71 ± 0.29 [b] | 25.89 ± 0.36 [b] | 3.40 ± 0.25 [a] | 44.28 ± 1.19 [b] | 8.34 ± 0.64 [c] | 312.46 ± 3.61 [b] |
| 60 | 6.07 ± 0.09 | 12.68 ± 0.39 [a] | 25.53 ± 1.04 [b] | 3.22 ± 0.43 [a] | 43.98 ± 2.31 [b] | 8.85 ± 1.04 [bc] | 312.22 ± 2.51 [b] |
| 80 | 6.14 ± 0.10 | 12.88 ± 0.86 [a] | 25.03 ± 0.34 [b] | 3.17 ± 0.46 [a] | 45.59 ± 1.85 [b] | 9.03 ± 0.52 [bc] | 309.08 ± 3.32 [b] |
| 100 | 6.17 ± 0.15 | 12.76 ± 0.62 [a] | 24.99 ± 0.52 [b] | 3.03 ± 0.59 [a] | 46.10 ± 2.24 [b] | 9.67 ± 0.50 [b] | 311.96 ± 2.94 [b] |
| F-test | ns | ** | ** | * | ** | ** | ** |
| C.V.% | 1.82 | 3.22 | 4.06 | 2.28 | 3.62 | 6.87 | 1.05 |

Data are represented as mean ± SD (*n* = 5). Different letters in the same column indicate significant differences between the treatments according to DMRT at $p < 0.05$. ** There were significant differences at $p < 0.01$, * There were significant differences at $p < 0.05$ and ns = not significant.

Interestingly, carbohydrates were the most abundant macronutrients and the highest levels between 40.98 and 50.40%. However, adding coconut water resulted in a significant decrease in the carbohydrate content produced by the fruit bodies of *H. erinaceus.* Similarly, the dietary fiber after exposure to coconut water was found to be decrease significantly, with the dietary fiber content between 7.95 and 9.67% (Table 4). The results show that the carbohydrate and dietary fiber content analysis of *H. erinaceus* tends to be in the same direction as that of the protein content as the increased crude protein and ash content causes the degradation of carbohydrates and fiber during the maturation process, thereby reducing their content [45]. The amount of carbohydrates and dietary fiber of *H. erinaceus* decreases in the same direction because dietary fiber is a compound polysaccharide, most of them are complex carbohydrates. Similarly, the energy value of *H. erinaceus* after exposure to coconut water was significantly decreased. The energy value varied significantly between treatments, from 301.65 to 318.30 kcal. The highest energy value was obtained from the control treatment, with an energy value of approximately 318.30 kcal (Table 4). The determining of the energy content depends on the components of *H. erinaceus* that provide energy are carbohydrates, fat, and protein. These are the main building blocks of energy-yielding nutrients, when combined with all components representing the nutritional energy value of *H. erinaceus*, it was found to be less than the control treatment. The energy conversion factors were a result of the coconut water administration, which resulted in a lower percentage of carbohydrates of *H. erinaceus*, which is the main component of energy. Nevertheless, the findings of the study provide a challenge in terms of comparing them with existing literature. To the best of our knowledge, this study represents the initial investigation into the effects of coconut water administration on the nutritional composition of *H. erinaceus.*

### 3.5. Analysis of Total Triterpenoid Content of H. erinaceus

The analysis of total triterpenoid content found in ethanolic extracts from *H. erinaceus* is presented in Figure 3. The results revealed that the coconut water resulted in a statistically significant increase in the total triterpenoids content of *H. erinaceus*. The coconut water at a concentration of 20–100% (*v/v*) resulted in the highest total triterpenoid contents from 67.87 to 89.24 mg Urs/g DW, whereas the triterpenoids in the control treatments were 56.84 mg Urs/g DW, indicating that coconut water contains phytohormones that act to induce the most significant increase in total triterpenoid synthesis at this concentration. As mentioned above, it has been shown that components in coconut water have a broad spectrum of physiological effects. In addition to regulating plant growth and development [46,47], coconut water also participates in stimulating the biosynthesis of triterpenoids. However, this study represents the first research concerning the effect of coconut water on concerning triterpenoid production and the factors that the factor that

increase the triterpenoid content of *H. erinaceus*. This is the result of adding coconut water, which is qualified as a phytohormone that exerts various roles in the different aspects of growth and development of *H. erinaceus*. Also included are some hypotheses that have also been presented to explain the potential trade-off between the synthesis of secondary metabolites, according to previous research conducted by xu et al. [46], which investigated the feasibility of enhancing triterpenoid production of *Inonotus obliquus* by adding methyl jasmonate (MeJA) as an exogenous elicitor agent, and it was found that total triterpenoid production increased by 53.6% compared with the control. This may indicate that the presence of phytohormone receptors in fungi revealed that fungal Histidine kinases can be an effective stimulant of triterpenoid biosynthesis from phytohormones in coconut water for mushroom cultivation [47]. In general, plants' triterpenoid biosynthesis is derived from the cyclization of 2,3-oxidosqualene, which in turn is derived from isopentenyl pyrophosphate generated through the mevalonate pathway [48,49]. According to research by Moses et al. [50], it was determined whether expression of the 2,3-oxidosqualene was affected by phytohormonal related to triterpenoid biosynthesis. Adding phytohormones also increased the expression of the triterpenoid genes to different hormone treatments, which further supports the existence of different spatiotemporal regulatory cues. In this study, triterpenoid genes were found to be induced by cytokinin, and gibberellin, which is a result of both types of hormones induced 2,3-oxidosqualene expression by 2- to 3-fold, which correlates well with the respective enrichment of the triterpenoid. Therefore, the results of such experimental results provide new insight into the mechanism by which phytohormones in coconut water act to increase triterpenoid yield in *H. erinaceus*.

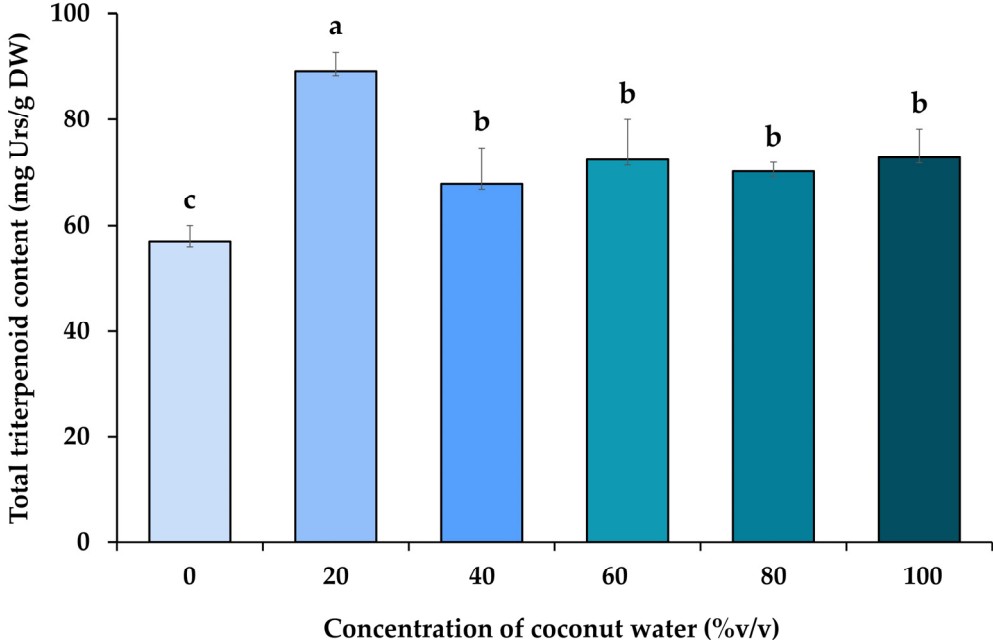

**Figure 3.** Total triterpenoid content of *H. erinaceus* from the exogenous application of coconut water at various concentrations. Values are represented as mean $\pm$ SD ($n = 5$). The different letters in the same column indicate significant differences between the treatments according to DMRT at $p < 0.05$.

### 3.6. Analyses of Total Phenolic Content of H. erinaceus

The results revealed that the coconut water resulted in a statistically significant increase in the total phenolic content of *H. erinaceus*. The coconut water at a concentration of 20–100% (*v/v*) improved the synthesis of phenolic compounds resulting in total phenolic content in the range of 15.90–17.39 mg GAE/g DW, while the control treatment had total phenolic content equal to 14.42 mg GAE/g DW (Table 5). The results demonstrated the potential of the tested coconut water as a viable elicitor that can effectively improve the production of bioactive compounds in *H. erinaceus*. The favorable effect of coconut water was partly due

to the presence of auxin, cytokinin, and gibberellins in the phytohormone, which was an important stimulant for the biosynthesis of phenolic compounds. According to research by Chaurasia et al. [51], the auxins and gibberellins at specific concentrations enhanced the growth and accumulation of total phenolic content in *Pleurotus florida* with an increase from control by 7.17 and 4.76%, respectively. In nature, the production of secondary metabolites from the phenylpropanoid pathway in fungi is the defensive response to environmental stimulus [52–54]. Upon activation by signal molecules, the cellular defense mechanisms located on the plasma membrane surface will combine with the signal molecules for the transcription of defense genes [54,55], and the accumulation of secondary metabolites through oxidative catabolism during fungal interaction or elicitor treatment [54,56]. The result is in the same direction that the presence of auxin and gibberellins in the phytohormone strongly influenced the biosynthesis of secondary metabolites in mushroom. Overall, the study demonstrated the ability of coconut water to serve as a suitable stimulus for the production of phenolic compounds in *H. erinaceus*. Even though the underlying mechanism of coconut water regulating plant secondary metabolic biosynthesis pathway has yet to be elucidated, the influence may be related to the presence of auxin and cytokinin in the phytohormone. Therefore, the relationship of coconut water in the secondary metabolite biosynthesis pathway needs further studies.

**Table 5.** Total phenolic content and DPPH scavenging activity of *H. erinaceus* from various concentrations of coconut water treatment.

| Concentration (%$v/v$) | Total Phenolic Content (mg GAE/g DW) | DPPH Scavenging Activity (IC$_{50}$, mg/mL) |
|---|---|---|
| 0 | $14.42 \pm 5.26$ [c] | $0.77 \pm 0.10$ [c] |
| 20 | $17.07 \pm 9.57$ [a] | $0.60 \pm 0.03$ [a] |
| 40 | $15.90 \pm 9.98$ [b] | $0.69 \pm 0.05$ [b] |
| 60 | $16.62 \pm 9.86$ [ab] | $0.64 \pm 0.01$ [ab] |
| 80 | $16.80 \pm 3.84$ [ab] | $0.64 \pm 0.04$ [ab] |
| 100 | $17.39 \pm 4.55$ [a] | $0.58 \pm 0.01$ [a] |
| F-test | ** | ** |
| C.V.% | 7.27 | 5.58 |

Data are represented as mean $\pm$ SD ($n = 5$). Different letters in the same column indicate significant differences between the treatments according to DMRT at $p < 0.05$. ** There were significant differences at $p < 0.01$.

The most attractive properties of bioactive compounds have steadily increased in recent years due to the demand for products with a direct effect on human health. Among the existing natural antioxidants in mushroom extracts, the phenolic compounds are known to exert a direct inhibitory effect on free radicals [57,58]. The effect is manifested through their chemical structure, in particular by the number of hydroxyl groups, which is directly proportional to antioxidant activity, and by their location in the molecule [57,59].

*3.7. DPPH Radical Scavenging Activity*

The assay on DPPH radical is useful in predicting antioxidant activities by inhibiting lipid oxidation. Thus, DPPH radical scavenging determines the extent to which free radicals are scavenged [60]. The antioxidant properties of ethanolic extracts from *H. erinaceus* assayed by the DPPH radical scavenging method were compared with standardized BHT (IC$_{50}$ value of 14.08 µg/mL).

Results from various studies vary and this may be due to different concentrations of coconut water. The DPPH radical scavenging activity of *H. erinaceus* extracts is shown in Table 5. The *H. erinaceus* extract after treatment with coconut water showed significantly higher antioxidant activity. Specifically, the coconut water treatments, ranging from 20–100% ($v/v$) concentrations, exhibited the most potent antioxidant activity, as shown by IC$_{50}$ values ranging from 0.58 to 0.69 mg/mL were significantly more effective at scavenging DPPH radicals than the control treatment with the IC$_{50}$ value of 0.77 mg/mL. The

enhanced antioxidant activity seen in this study can be related to the stimulating impact of the coconut water treatments on the phenolic content in *H. erinaceus*. In contrast, the control treatment exhibited decreased antioxidant activity due to the limited synthesis of phenolic compounds. Moreover, it has been established that phenolics have a direct correlation with several biological activities, including antioxidant activity [61–63]. The theory presented in this study is supported by empirical data indicating that the administration of exogenous coconut water leads to the upregulation of secondary metabolites. Particularly, it has been observed that coconut water serves as a major stimulator in the production of triterpenoids and the accumulation of phenolic compounds. The present results suggest that coconut water has a notable impact on bioactive compound levels, particularly when interacting with phytohormones on bioactive compounds levels, which will affect antioxidant activity, and the research of Aremu et al. [64] has shown that the production of higher content of phytochemicals with the appropriate cytokinin might be the key reason for the higher capacity to remove free radicals content. In another study, Yuan et al. [54] conducted tests using methyl jasmonate-treated cultures of *Sanghuangporus vaninii*, showed higher activity in scavenging DPPH free radicals than those found in the control treatment. However, no significant pattern was established between the quantified phenolic compounds and the antioxidant activity of the *H. erinaceus* extracts. We assumed that the interaction among the various identified compounds produced complex and interwoven responses, which will require further study for better elucidation.

## 4. Conclusions

This study revealed the effects of coconut water on phytohormones such as auxins, cytokinins, and gibberellins, proving the feasibility of applying coconut water to increase production, nutritional value, bioactive compounds, and antioxidant activity of *H. erinaceus*. The application of coconut water at a concentration of 20% ($v/v$) resulted in a significant enhancement in the development and production of *H. erinaceus*, in comparison to the control treatment. In a similar vein, the nutritional composition of *H. erinaceus* was influenced by the application of various concentrations of coconut water, resulting in changes in ash content, crude protein, and crude fat. However, the utilization of coconut water at a concentration of 20% resulted in higher crude protein. In addition to the aforementioned characteristics, edible and medicinal mushrooms are increasingly acknowledged as an untapped source of diverse bioactive compounds that possess significant health-enhancing benefits of *H. erinaceus*. The administration of various amounts of coconut water led to the stimulation of triterpenoid and phenolic compound production, hence encouraging increased antioxidant activity. Both categories of bioactive compounds are classifications of naturally occurring substances that possess notable bioactive properties frequently seen in medicinal mushrooms. Further studies should be directed to the growth mechanisms and various synthetic processes of *H. erinaceus* under exogenous coconut water application. However, our study revealed for the first time the coordination in the application of coconut water to the production of *H. erinaceus* and provided information about the development of new mushroom cultivation methods. Moreover, coconut water, with its many applications, is one of the most important natural products that are safe for users and influence environmental sustainability.

**Author Contributions:** Conceptualization, P.C. (Preuk Chutimanukul) and O.P.; methodology, P.C. (Preuk Chutimanukul), O.P. and S.S.; validation, P.C. (Preuk Chutimanukul), S.S. and O.P.; formal analysis, P.C. (Preuk Chutimanukul), O.T., S.P. and P.C. (Panita Chutimanukul); investigation, P.C. (Preuk Chutimanukul), O.P. and S.S.; resources, P.C. (Preuk Chutimanukul), O.P. and S.S.; data curation, P.C. (Preuk Chutimanukul), H.E. and P.C. (Panita Chutimanukul); writing—original draft preparation, P.C. (Preuk Chutimanukul), O.P. and O.T.; writing—review and editing, P.C. (Preuk Chutimanukul), O.P. and S.P.; supervision, P.C. (Preuk Chutimanukul); project administration, P.C. (Preuk Chutimanukul); funding acquisition, P.C. (Preuk Chutimanukul). All authors have read and agreed to the published version of the manuscript.

**Funding:** This research was funded by Thammasat University Research Fund, Contract No. TUFF 06/2565.

**Institutional Review Board Statement:** Not applicable.

**Informed Consent Statement:** Not applicable.

**Data Availability Statement:** The original contributions and data presented in this research are included in the article; further inquiries can be directed to the corresponding authors.

**Acknowledgments:** The authors would like to thank Thammasat University Center of Excellence in Agriculture Innovation Centre through Supply Chain and Value Chain, Faculty of Science and Technology, Faculty of Medicine, Thammasat University for providing technical supports and instrument. We are also grateful to Fresh & Friendly Farm Co., Ltd., Thailand, for providing mushroom production place and instrument. We would like to thank Green Spot Co., Ltd., Thailand, for materials.

**Conflicts of Interest:** The authors declare no conflict of interest. The funders had no role in the research in the analyses, or interpretation of data; in the writing of the manuscript; or in the decision to publish the results. This research was conducted in the absence of any commercial relationships that could be construed as a potential conflict of interest.

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
