# Peer review of "Exogenous Application of Coconut Water to Promote Growth and Increase the Yield, Bioactive Compounds, and Antioxidant Activity for Hericium erinaceus Cultivation"

_horticulturae, doi:10.3390/horticulturae9101131_

Round 1

Reviewer 1 Report

Dear authors,

Thank you for the manuscript entitled "Optimization of Hericium erinaceus cultivated by exogenous application of coconut water to promote growth and increase the yield, bioactive compounds, and antioxidant activity". My comments are as follows:

English can be improved. Please re-read again or maybe can get it from a English native speaker or submit it for proofreading.

Abstract: Please add the research problem and simple methodology in your abstract. It would give better clarity tho the readers

Line 24: 22.09% increase-  increase of what? growth? yield? weight? please specify

Line 29, 31 and few other Lines: change treatment control to control treatment.

Line 30: every concentration. Do you mean all concentrations?

Introduction

Line 44: after health benefit no need to put '.' Check for the rest of the manuscript for the same error.

Line 44: At now? maybe can change to 'Currently' for better choice of word

Line 47/66: italic the scientific name.

Line 69: what do you mean with contemporary times?

Line 69-79-  no reference for this section.

Line 81: Specific substances? such as?

The authors suggest the use of coconut water in the cultivation of H. erinaceus, however, the authors did not include what others have been used to cultivate H. erinaceus, thus, bringing up the question of why we need to add something to the original media. 

At the same time, an explanation of proximate analysis, triterpenoid, phenolic, and antioxidant was not found in the introduction which is important since it is the main part of the study.

Methodology

Line 92: capital for Kjedahl

Line 111: 15 psi-  use SI unitLIne 132 wrong way of citation

Line 148: 4 ml spray. If you are using the spray, how do we know we spray 4 ml on each? This is very important so people can replicate the study.

Line 155: fresh of H. erinaceus? what do you mean here?

Line 175: sentence seems hanging.

Line 178: carbs?

Line 181: The first sentence missing something. The sentence is not complete.

Line 186: do you need to use bracket here?

Line 189: at the conclusion of the experiment?. No full stop after experiment

Line 220/222- italic scientific name.

Line 224- correct the concentration of DPPH

Results and discussion

Line 244: pH 7.77? are you sure?

Line 251: very hard to understand the sentence

Line 268: Capital H on However

Line 271-273. Check again the sentence

Line 287: MIssing fulls stop before It was found?

Line 441: coconut- no need capital letter

LIne 477. Re write the sentence

Conclusions

Need to re-do the conclusion by stating the best treatment for each of the parameters in the experiment. There is no data presented in the conclusion hence it is very hard to say which treatment are the best.

References

PLease follow the format for Horticulturae

Some error found like:

Line 547: no journal mentioned

Line 559/576/579: all caps

English is OK

Reviewer 2 Report

Comments on the ms. horticulturae-2613332 entitled “Optimization of Hericium erinaceus cultivated by exogenous application of coconut water to promote growth and increase the yield, bioactive compounds, and antioxidant activity”

Overall

The topic of the manuscript is of high interest to the international readership. The selection of an appropriate cultivation method of Hericium erinaceus is a highly important issue that will improve the yield, productivity and active compounds concentration in this edible mushroom. The main objective of this research was to examinated the possible advantages of administering coconut water in order to augment the yield, bioactive chemical content, and antioxidant activity of H. erinaceus and the improvement of other mushroom cultivation methods.

I have a few comments and factual inquiries:

Firstly, the chemical composition and phytohormone content of the coconut water used in the experiment were not determined. Only the dilution of the coconut water sprayed on the fruit bodies was reported.

Without knowing the composition of the coconut water, the experiment is not reproducible because we do not know the specific content of phytohormones.

In this situation, there is no certainty that another ingredient in the coconut water did not influence the results.

In their conclusions and discussion, the authors attribute a stimulatory role to coconut water at appropriate concentrations in increasing the yield and medicinal properties of the cultivated mushroom, so this is a key issue.

In support of the stimulating properties of coconut water, previously published papers are cited, but these concern plants, not fungi eg. 39-41, 58-62 in „References chapter”.

So I wonder if the conclusions drawn are not too far-fetched. In the peer-reviewed article, please cite the articles that deal with fungi stimulated by plant phytohormones.

Finally, I think that the experiment should be repeated, but I do not know if this is possible because the results obtained are not reproducible and it is impossible to determine which factor influenced them.

I don't have any major comments, I did pay attention to minor faults in the formatting of the text.

Reviewer 3 Report

The authors have designed a very simple but interesting study to investigate the use of coconut water as an elicitor for enhancing the yield, nutrient content, and antioxidant activity of the edible fungus Hericium erinaceus. Overall, the study is well-structured and compelling, but there are a few minor issues that need attention. My main concerns apply to the use of English and the simplicity of the discussion. I recommend revising the English language and enriching the discussion with more relevant bibliography to support and contextualize the obtained results. Additionally, it is worth noting that the authors heavily rely on the phytohormone content of coconut water to explain their findings. Since this content has not been thoroughly studied, it remains uncertain whether it is the primary reason for the observed significant differences. It would be beneficial to consider conducting an analysis of this aspect for each water treatment.

Here are the specific corrections:

Line 38: "Mushrooms are generally consumed [...]".

Line 74: "It may be more pertinent to explain the significance of phytohormone-like compounds in fungal metabolism, rather than in plants, as it is irrelevant in this context."

Line 244: "The text mentions a pH of 7.77, while Table 1 reports a pH of 6.77.

I think the English should be revised before publishing the manuscript.

Reviewer 4 Report

Manuscript ID: Horticulturae-2613332

Title: Optimization of Hericium erinaceus cultivated by exogenous application of coconut water to promote growth and increase the yield, bioactive compounds, and antioxidant activity
Journal: Horticulturae

Submitted to section: Medicinals, Herbs, and Specialty Crops; The Edible Mushroom Industry: A Vital Component in Horticultural Production

The submitted article explores how the addition of coconnut water can lead to improved productivity and several bioactive metabolites amounts in Hericium erinaceus mushroom, widely used for direct consuption and medicinal purposes. The topic is interesting and, from what is depicted in the article, the demanding experimental work carried out can be guessed. Unfortunately, the manner in which it is described in this article is confusing, repetitive, unclear in many places (many sentences are incomplete and therefore incomprehensible) and superficial. Added to this are the numerous grammatical and lexical imperfections and in the construction of English sentences, present throughout the article, so much so that in many parts it makes for heavy reading and complicated interpretation of what the authors report.

Below are many of the points that need to be changed, but I recommend that the text be completely revised, perhaps with the help of native English expert.

Title

Lines 2-4: In my opinion the title should be changed: "Optimization of Hericium erinaceus..." optimise what?

Abstract

The abstract is missing a brief overview of the experimental plan.
Lines 21-25: these sentences are repetitive.
Lines 23 e 26:
"..concentration.." you must always specify % (v/v); "..levels of protein assessed by nutricional analysis..." it is not correct to speak of nutricional analysis
Line 27: "..
content.." must be entered before "..as triterpenes..."
Line 32-34: the sentence appears to be more of the introduction rather than the conclusion and therefore reiterates what has been written previously.

Introduction

Although the topic seems to be sufficiently introduced, in many places the paragraph is repetitive, superficial and in some places unintelligible. It is necessary to supplement the introduction with information related to the productivity of these cultivars, the use of alternative substrates for improvement, and the environmental sustainability and circular economy implications that justify their use. It is also necessary to give more information about similar studies reported in the literature on the use of coconut water as a growth substrate supplement.

Materials and Methods

The Materials and methods section also needs to be improved and supplemented. There is no uniformity of information on the analyses carried out, some being described accurately, others superficially and others not described at all even if reported. (e.g., those on coconut water). For each matrix analyzed, the sample homogenization process is not accurately described, it is not specified what the analyzed sample consists of, nor how many analytical replicates are performed. The number of experimental replicates (how many bottles for each treatment) is not specified. The location of the experiment is not accurately described.

Chemicals and Reagents

Some chemical formulae are not accurately written. However, I don't think it is necessary to include both the chemical name and the formula of the reagents used.
Line 92: "...
kyeldahl copper...." perhaps "kyeldahl copper catalyst tablet"
Line 97: "...
vanadate-molybdate reagent were ....Ricca Chemicals...", previously (lines 93-94) listed the individual components of this reagent (ammonium molybdate and ammonium metavanadate) but of another brand (Kem House). Why?

Line 98: the degree of purity of the reagents and standards named in that sentence is not reported.

Sample Preparation
I would title this paragraph "Experimental plan or design" rather than "Sample preparation" and integrate it with the information in 2.4 concerning the production of substrates supplemented with Coconut water.
Line 106-107: more information should be provided to better identify the site where the experiment was conducted; what is the % composition of the components of initial cultivation substrate? how is it prepared and homogenized?
How many experimental replications are carried out for each of the substrates used (control and the 5 substrates enriched with coconut water)?
Analyses of the physicochemical characteristics ......
Line 121: " ...by subjecting them to substrate assessment ..." remove as repetitive and unnecessary. Line 121-130: the whole period is too long and is poorly structured. It is recommended to break it up into several sentences.
Lines 131-132:
"..on the approach developed by Udelhoven et al. for the analysis of nutrient content ..." Reference 23 does not agree with the sentence to which it refers; in this paper, analytical data on soils obtained by means of bidirectional diffuse reflectance measurements are presented and correlated with measurements performed with conventional methods for which the relevant references are found in the paper. So, what is "a modified method based on approach ......?
Lines 133-137: The description of the methods is poor and needs to be improved. Several reagents listed in section 2.1, which are known to be used for this type of analysis, are never mentioned in the article for this very reason. I therefore wonder what the purpose of such a meticulous list is.
It is important also to specify how many analytical replicates are made for each type of analysis.

Study of Growth and Yeld...
Line118-119: as already mentioned, the whole of paragraph 2.4 should in my opinion be integrated into 2.2 by giving it the more appropriate title specified before.
Line 143-144: what analyses were carried out on coconut water? the text is missing a table with the results of these analyses, nor are the analytical methods indicated.
Lines 149-150: this sentence is unclear; however, I would change "...recorded..." to "...characterized...".
Lines 154-157: this sentence should be amended and clarified.
Proximate composition of H.erinaceus
What is Proximate composition?
Line 163: How is each sample submitted for analysis composed? better clarify.
Line 165: " ...
to facilitate..." is not a suitable term; rather than nutritional anaysis I would speak of analysis of nutrients.
Line 168: change
"....(analyze the total amount of nitrogen and multiply by 4.38)...." to "(=Total nitrogen x 3,48 (add reference))"
Line 169: was an analyser used for the macro-Kjeldhal method? which one?
Line 170: delete
"...to protein quantification."
Line 171: what volume of petroleum ether per gram of sample?
Line 175: the sentence is incomplete and therefore unclear; however, the whole description of this method is unclear and needs to be improved.
Line 179: reference 25 does not agree with the sentence to which it refers; in this paper there is no trace of equation (3)

Preparation of H. erinaceus ethanol extract

Lines 181-182: the sentence is incomplete. The sample preparation procedure is different from that illustrated in lines 163-164; why is that?
Line 186:
"(The process...)" the sentence should not be bracketed and rewritten more clearly.
Line 188: "
decanted" is not the right word. Perhaps "dried under vacuum" was the correct term? Total Triterpenoid content...

Line 196: "....w/v..." mg/ml? or? better to specify.
Total Phenolic contents...
Line 208-209: "by" is repeated twice in a row. Improve the sentence.
Line 209-212: usually the method involves a dilution of the sample to be analyzed with water followed by the addition of the Folin-Ciocalteu's reagent and finally the sodium carbonate solution to adjust the pH. As it is written it seems that water is not added and the carbonate is placed before and after the dye. Why?
Statistical analysis
The paragraph should be integrated with the information concerning the number of experimental replicates and the number of analytical replicates never specified in the previous paragraphs.

Results and Discussion

The general considerations highlighted for the other chapters can also be extended to Results and Discussion. It is unconnected, in some places one does not understand the meaning or usefulness of what is written, and again, there are many incomplete sentences and others that are poorly constructed with bad style and form and therefore difficult to read and understand.

So, it needs to be completely rewritten

Analyses of the physicochemical.....

Line 243-249: all the reported talk about the pH of the starting substrate (in brackets in table 1 it is 6.77 and not 7.77 as in the text) is unjustified and unclear. It is normal that a starting substrate with a pH suitable for the development of this species of mushrooms was chosen.
Line 249-276: naturally the same can also be extrapolated to the remaining nutrients analyzed.

Study of growth and ......

Line 278-280: the sentence is obvious.
Line 280: 282: why is the verb in the future tense?
Line 282-283: incomplete sentence.
Line 290-292: poorly written sentence: "which" is repeated twice in a row.
Lines 293-296: to be rewritten, repetitive.
Line 297-303: ripetitive sentences
Line 311- 334: sentences written with bad style and form and therefore difficult to read and understand.
Proximate composition.....
What is Proximate composition?
Line 336-394 : the sentences in whole paragraph are written with bad style and form and therefore difficult to read and understand.
Analysis of total triterpenoid ......
Line 396-437: the sentences are written with bad style and form and therefore difficult to read and understand.
Analyses of total phenolic content.... and DPPH radical scavenging activity
Lines 438-475: the observations highlighted above also apply in this case

Numerous grammatical and lexical imperfections and in the construction of English sentences are present. There are many incomplete sentences, also, and others that are poorly constructed with bad style and form and therefore difficult to read and understand.

Round 2

Reviewer 2 Report

I thank the Authors of the paper for the corrections made, which clearly improved the value of the paper, the interpretation of the results, and its subsequent use.

I have no major comments on the quality of the language used in the work. However, please read the text carefully to remove minor spelling errors.

Reviewer 3 Report

The authors have once again presented a very simple but interesting study, aimed at investigating the potential use of coconut water as an elicitor to enhance the yield, nutrient content, and antioxidant activity of the edible fungus Hericium erinaceus. Firstly, I appreciate the authors for the improvements they have made to the manuscript, particularly the inclusion of the analysis of the substrate and the phytohormone content of the coconut water. Both of these additions were essential for the manuscript's suitability. However, the primary issues that concern me remain unchanged.

One major concern is the deteriorating quality of English in this version compared to the initial manuscript. It is riddled with sentences that lack coherence, as well as numerous orthographical and grammatical errors. Furthermore, the discussion section remains almost non-existent, relying heavily on repetitive mentions of phytohormone content as the sole key factor for any significant results. To enhance the manuscript's quality, I strongly recommend thorough language revision and an enriched discussion section supported by relevant bibliography to provide context and support for the obtained results.

Additionally, it would be highly beneficial for the authors to delve into explaining the differences among treatments with coconut water and why the best result was achieved with the minimum content of it. I maintain my stance that the manuscript requires substantial improvement before it can be considered for publication.

Minor Issues:

  • Line 19: The sentence is unclear and lacks coherence.
  • Line 68: The sentence lacks clarity and coherence.
  • Line 292: The content of Table 1 and its related information should be included in the Materials and Methods section as preliminary information preceding the experiment itself.
  • Line 300: Similarly, the discussion of the phytohormone content of the coconut water should have been considered before conducting the assays and should not be treated as a result of the study.
  • Line 317: Information related to Histidine kinases, as presented here, should be incorporated into the introduction section. The authors merely present this information without engaging in meaningful discussion.
  • Line 432: The sentence is unclear and lacks coherence."

Extensive editing of English language required

Reviewer 4 Report

The revised manuscript has been extensively edited by the authors, so the effort produced can only be appreciated. Unfortunately, however, these changes are not sufficient to publish the article as presented to us. In many places lexical and English sentence construction imperfections and orthographical and grammatical errors remain. The revised manuscript is confusing, repetitive, incoherent, unclear in many places (many sentences are incomplete and therefore incomprehensible). Newly added parts also require further corrections and modifications. On the other hand, it can be intuited that the experiments were conducted correctly and the results are significant, so from this point of view, I think it is worthy of publication following, however, a thorough revision from a lexical point of view, perhaps with the help of native English experts.

That said, it is necessary to change the following points:

The sentences at Lines 19, 62, 68-78, 82-88, 190-193, 226-227, 297-299, 304-306, 317-329, 333-365, 371, 376-380, 384-385, 417-419, 423-426, 432-466, 473-475, 480-484, 487-491, 501-503, 515-516, 520-534 must be edited because they are unclear and/or incomplete and/or inconsistent, full of grammatical and lexical errors and poorly constructed.

Reference 24 and 25 do not seem to fit the claims for which they are mentioned. I think there has been a swap.

Line 158: It is important that there has been this addition in the text, but it is necessary to supplement the added information: firstly, the make and model of the CE-MS-MS used is missing, and, secondly, it is good to describe, even briefly, the method used, even if a reference is given for this

Answer 26 is not clear. I assume that all parameter data analyzed are calculated on the dry weight of the sample. Therefore, what is the optimum time to completely dehydrate such samples 48 or 72h? To overcome the reported problem of stability of some of these parameters, have other methods such as freeze-drying not been considered?

Lines 379-380: what does the number 9.31 % refer to? In tab. 4, there is no such value in the reference column.

Lines 411-412: probably the note refers to the parameter dietary fiber and not to ash content as reported.

Finally, it would be good for the authors to add their hypotheses regarding the fact that the maximum effect on some parameters is found with the addition of a minimum dose of coconut water.

In many places they are present lexical and English sentence construction imperfections and orthographical and grammatical errors.  It is necessary that the manuscript is completely reviewed by native English speaking experts.
